# Toxic Effects of Cadmium on Fish

**DOI:** 10.3390/toxics10100622

**Published:** 2022-10-19

**Authors:** Yinai Liu, Qianqian Chen, Yaoqi Li, Liuliu Bi, Libo Jin, Renyi Peng

**Affiliations:** Institute of Life Sciences & Biomedicine Collaborative Innovation Center of Zhejiang Province, College of Life and Environmental Science, Wenzhou University, Wenzhou 325035, China

**Keywords:** aquatic ecosystems, food chain, cadmium accumulation, cadmium toxicity, oxidative stress

## Abstract

Large amounts of enriched cadmium (Cd) in the environment seriously threatens the healthy and sustainable development of the aquaculture industry and greatly restricts the development of the food processing industry. Studying the distribution and toxic effects of Cd in fish, as well as the possible toxic effects of Cd on the human body, is very significant. A large number of studies have shown that the accumulation and distribution of Cd in fish are biologically specific, cause tissue differences, and seriously damage the integrity of tissue structure and function, the antioxidant defense system, the reproductive regulation system, and the immune system. The physiological, biochemical, enzyme, molecular, and gene expression levels change with different concentrations and times of Cd exposure, and these changes are closely related to the target sites of Cd action and tissues in fish. Therefore, the toxic effects of Cd on fish occur with multiple tissues, systems, and levels.

## 1. Introduction

Cadmium (Cd), a toxic and nonessential transition metal, is a major threat to common organisms in aquatic ecosystems and has been widely recognized as a major pollutant in waters worldwide, and Cd travels up the food chain, eventually threatening human health, thus, Cd poses health risks to both humans and animals [1,2,3,4]. Cd causes a wide range of toxicological effects, including stomach cramps, vomiting, diarrhea, hematuria, kidney failure, and other symptoms. The Joint Expert Committee on Food Additives (JECFA) has stipulated that the allowable weekly intake limit of Cd for humans is 7 μg/kg, and exceeding the limit harms human health [1,5]. Epidemiological data suggest that environmental Cd exposure may be associated with various types of cancer and is also an important factor in osteoporosis [6,7]. Moreover, the liver and kidney are extremely sensitive to Cd toxicity, probably due to the ability of these tissues to synthesize metallothionein (MT), which is a Cd-induced protein that protects cells by tightly binding toxic Cd ions, thereby resulting in oxidative stress, this may be a cause of several liver and kidney diseases [8].

Cd has several characteristics, including high environmental toxicity, easy accumulation, and poor degradation. For organisms, Cd displays toxic effects on many organs, and Cd pollution is not easily found, however, Cd is easily enriched and can transmit its toxic effects through the food chain [9]. Cd pollution in the environment is caused by the development and utilization of chemical fertilizers in agriculture, metals and minerals in industry, and the use of some industrial products containing Cd, which may be discharged into water with wastewater. Even if the concentration of Cd is sometimes low, it can accumulate in algae and sediment, be absorbed by fish, shellfish, shrimp, and crabs in water, be concentrated through the food chain, and eventually enter the human body [1,10,11,12].

As fish are among the most important sources of food protein for human beings, it is highly likely that Cd can be enriched in fish and transferred into the human body through the food chain. Cd accumulation causes damage to the cardiovascular, immune, reproductive, and nervous systems in the body [13]. At the same time, it disrupts the cell cycle, proliferation, differentiation, and DNA replication and repair at the cellular level and then affects the apoptosis pathway [14,15]. Fish are an essential group of organisms in the aquatic ecosystem that are at the top of the food chain and are very sensitive to Cd pollution. If fish are exposed to Cd pollution for a long time, their different tissues will absorb and accumulate Cd, thus affecting the structure and function of the gills, liver, and gonad. Furthermore, Cd affects their physiological metabolism and reproductive and immune systems and eventually leads to metabolic disorders, physical disorders, or even death [16,17,18]. Recently, studies have shown that Cd can induce various epigenetic changes, thus, changes occur in the chemical modification of DNA, histones, and chromatin, but not the sequence of DNA nucleotides, including DNA methyltransferase, histone acetyltransferase, histone deacetylase, histone methyltransferase, and microRNA. As a result, fish incur a greater risk of disease and development of various types of cancer [18,19,20,21]. Therefore, using these indicators to indicate the impact that Cd pollutants have on the environment is an effective way to evaluate the safety of the ecosystem.

The toxic effects of Cd exposure in different fish are not the same. Adult male and female *Japanese medaka* were exposed to the non-lethal consequences of Cd, with 0–10 ppb concentration for 7 weeks, and the results displayed that gonadal steroid release was significantly decreased, while female plasma estradiol levels altered more significantly. Meanwhile, physiological response to Cd exposure along the hypothalamus–pituitary–gonadal axis, including the hepatic estrogen receptor, plasma vitellogenin, plasma steroids and gonadal–somatic indices, were more sensitive than the reproductive and developmental endpoints, and the reproductive and physiological endpoints for adult medaka pairs exposed to Cd for 7 weeks are shown in Table 1 [22].

When *Oreochromis aureus* were exposed to 0.5–10 ppm Cd, their food consumption showed a dramatic decrease, which would take 21 days to return to normal after fish were returned to Cd-free water, however, Cd was still in tissues for 34 days, and the accumulation was highest in the kidney after the fish were placed in Cd-free water [23]. Juvenile parrotfish, *Oplegnathus fasciatus*, were exposed to Cd mixed with a fish diet, and the result displayed that tissue Cd concentrations increased with the Cd content of the diets, and Cd accumulated the most in the liver, followed by the gill and then muscle [24]. Because zebrafish (*Danio rerio*) share 87% genetic homology with humans, their embryos, larvae, and adult fish are employed as standard models for many toxicity tests, and a summary of the toxic effects of cadmium in zebrafish is shown as Table 2 [25].

## 2. Detection Methods and Accumulation Effects of Cadmium in Fish

### 2.1. Cadmium Accumulation Induces Cytotoxicity in Fish

Studies have shown that Cd can enter cells through calcium ion channels. Cd and calcium (Ca) are both divalent metal ions. The radius of Cd^2+^ (0.097 nm) is similar to that of Ca^2+^ (0.099 nm), and Cd^2+^ has a higher affinity for anion binding sites in calcium ion channels than that of Ca^2+^, thus, Cd^2+^ can directly inhibit the active transport of Ca^2+^ for cells by competing with Ca^2+^. After Cd^2+^ enters cells, the physiological processes related to Ca^2+^ can be affected [46,47]. For example, Cd^2+^ can inhibit the outflow of Ca^2+^ by competing with Ca^2+^ for the binding site of Ca^2+^-ATPase, resulting in an increase in the intracellular Ca^2+^ concentration [48]. At the same time, Cd^2+^ can function as a substitute for Ca^2+^ to bind calmodulin (CaM), activate CaM-dependent kinases or directly activate its related enzymes, interfere with intracellular Ca^2+^-related signaling, and cause cytotoxicity [49,50]. In addition, Cd^2+^ can replace Ca^2+^ to bind to microtubules, microfilaments, and actin, destroy tight junctions and gap junctions between cells, damage cytoskeleton integrity, and ultimately affect the basic functions of cells [46,51,52]. Cd can induce cell apoptosis in two ways. First, Cd can bind to GSH rich in sulfhydryl groups on the mitochondrial membrane, causing proton (H^+^) influx, Ca^2+^ overload, and oxidative phosphoric acid uncoupling, which further leads to a decrease in the mitochondrial membrane potential, leads to the activation of Ca^2+^/Mg^2+^-dependent endonuclease (DNase I), and finally induces cell apoptosis [53,54]. Second, Cd can also lead to the release of cytochrome c and other apoptosis-related factors and increase the activity of caspases by inducing the opening of the mitochondrial permeability transition pore, which eventually leads to apoptosis [55,56]. In addition, Cd plays an important role in the induction of apoptosis by upregulating the expression of a series of proto-oncogenes [57]. The lethal effects of cadmium on different fish species are shown as Table 3.

### 2.2. Test Method for Cadmium Accumulation in Fish

Usually, there are multiple methods to detect the content of Cd in living organisms, and each method exhibits its own advantages and disadvantages. In the natural state, Cd mainly exists in the ionic state, but after entering the fish, most Cd ions quickly combine with metallothionein to form organic complexes, and a small part forms coordination compounds with nucleosides, proteins, and other substances in the body through ionic bonds [71,72]. At present, the methods used for detecting Cd in fish are usually atomic absorption spectroscopy (AAS) and inductively coupled plasma (ICP), the former includes graphite furnace atomic absorption spectrometry (GFAAS) and flame atomic absorption spectrometry (FAAS), and the latter includes inductively coupled plasma‒mass spectrometry (ICP‒MS) and inductively coupled plasma optical emission spectrometry (ICP‒OES) [73,74]. GFAAS has been widely used in the determination of Cd, due to its high sensitivity, low detection limit, and small sample amount, however, this method also has certain limitations, such as being susceptible to the influence of ash temperature, atomization temperature, matrix modifier condition, and the content of Cd in the sample. The FAAS method, which exhibits a high sensitivity, strong anti-interference ability, high precision, and good selectivity, is suitable for detecting the content of trace heavy metals, including Cd. The ICP‒MS method is widely used in determining the content of metal elements due to its advantages, including a strong anti-interference ability, low detection limit, high accuracy, good precision, convenient detection process, and accurate detection of low metal content. ICP‒OES has several advantages, including a wide linear range and a rapid and convenient detection process, but it is easily interfered with by spectrum and matrix effects [73,74,75].

### 2.3. The Accumulation of Cadmium in Fish Was Influenced by Feeding Habits and Routes

The degree of Cd accumulation in fish has a strong relationship with the fish feeding type. In the aquatic food chain, a step-by-step amplification effect, with increasing trophic levels, is observed for Cd accumulation. Considering the different cumulative effects of Cd on different feeding fish, the average accumulation coefficient of Cd in fish tissues can be generally arranged as omnivorous > carnivorous > herbivorous. The average accumulation coefficient of Cd, from largest to smallest, in the tissues of fish that live in different water layers is generally the bottom fish, middle and lower fish, and middle and upper fish, which may be due to a variety of factors, such as the position of the fish in the food chain and their metabolic capacity [76,77,78]. For instance, herbivorous fish mostly live in the upper and middle layers of water and mainly feed on phytoplankton or aquatic plants. The accumulation of Cd is mainly derived from intestinal food digestion. Omnivorous and carnivorous fish mostly inhabit the middle and lower layers of water and feed on small fish, aquatic insects, and shrimp, and they are exposed to Cd in the water environment and sediments, thus, the Cd accumulation in omnivorous and carnivorous fish is higher because they may ingest large amounts of Cd-containing sediments during feeding in the lower water [76,78]. 

The feeding route is another important factor that affects the accumulation and distribution of Cd in fish [79]. Cd is absorbed with water or foods and transported through the blood circulation to migrate and distribute in the different tissues and organs of fish. After being absorbed with water and transported through the blood circulation, Cd is mainly enriched in detoxifying organs, such as the kidney and liver, but when ingested with foods, Cd is mainly enriched in intestinal tissues, due to digestion and absorption in the intestine [80]. However, along with migration and distribution in fish, the accumulation of Cd that is both absorbed with water and foods displays gradual decreases in different organs or tissues. When absorbed with water, Cd ions are directly absorbed by gills in a dissolved state, and the accumulation of Cd in different tissues is generally in the following descending order: kidney, liver, intestine, gill, and muscle [81,82]. While absorbed with foods, they are absorbed by the digestive system, and the Cd accumulation in different tissues is generally in the following order, from high to low: intestine, kidney, liver, gill, and muscle [81,82,83]. The accumulation and distribution of Cd in fish were slightly different with different exposure methods. In order to assess health risk of some heavy metals, including Cd, from canned tuna and fish in Tijuana, Mexico, Rodriguez-Mendivil et al. diaplayed the range, mean concentrations, and incidence of occurrence in four species of fish for Cd in the analyzed samples in Table 4 [84].

## 3. Toxic Effects of Cadmium Accumulation on Fish

### 3.1. Cadmium Toxicity Damages Fish Tissue Structure

When fish are exposed to Cd for a long time, their organs or tissues become pathologically damaged, which can even lead to inflammation or necrosis [83]. The physiological functions of gills include regulating osmotic pressure, gas exchange, ion transport, acid-base balance, and excretion of metabolites, which play a key role in physiological, biochemical, and metabolic activities [85,86]. Gills are the first targets of Cd in water, through which fish continuously absorb Cd ions during respiration, and the absorbed Cd ions first accumulate in the surface cells of the gill tissue and are then transported through the blood to various organs or tissues. Therefore, Cd first has a toxic effect on the gill tissue structure of fish, leading to inflammation, apoptosis, necrosis, and so on [87,88]. Under Cd ion stress conditions in water, the structure of cells in gills changes, as well as the activities of ion transport and ATPase, thus, the ability of gills to regulate osmotic pressure and ion transport is affected. With the increase in concentration and time of Cd exposure, the problems, regarding inflammation, apoptosis, and necrosis, of gill tissue cells became more serious, and the lesions of the tissue structure became more obvious [89,90]. Evidence has shown that *Lates calcarifer* was exposed to 20.12 mg/L Cd for 48 h, resulting in aneurysm lesions in gill tissue. With the extension of exposure time, obvious pathological lesions, including hyperplasia, occurred in gill tissue at 72 h and 96 h [91]. The pathological changes in gill tissue caused by Cd mainly included cell proliferation, gill lamellae bending, aneurysm and cell shedding, and oxidative damage, such as vacuolization and swelling of mitochondria, and with the increase in Cd exposure dose and time, the problems became more obvious [88]. On the one hand, gill tissue lesions impact physiological function. On the other hand, the lesions also act a defense mechanism against Cd stress, as the distance that Cd must travel, the blood circulation is increased through the change in gill tissue structure, and to some extent, Cd can be prevented from entering other tissues in fish [92].

The liver, as the main detoxifying organ of fish, mainly protects the body from damage, turning heavy metals into less toxic substances, which are excreted from the body (Figure 1) [93]. Usually, Cd ions in the water environment first enter the fish body through gill respiration, then accumulate in the liver through the blood circulation system, and finally redistribute in various tissues. If fish are exposed to water that contains Cd ions for a long time, the Cd content in the liver exceeds its detoxification capacity, which affects the structural and functional integrity of fish liver tissue to different degrees [94,95]. With the increase in Cd concentration, the degree of irreversible damage to the structure of fish liver cells also increases significantly, which mainly includes endoplasmic reticulum expansion, nuclear deformation, nuclear membrane mitochondrial swelling, distortion, granular cytoplasm, and cell edema and cavitation. When the liver tissue structure is destroyed, the detoxification ability of the liver is significantly reduced, which has an irreversible impact on growth, reproduction, and survival [17,95,96].

### 3.2. Cadmium Toxicity Damages Fish Reproduction, Development and Endocrine System

Cd exposure seriously damages the reproductive system of fish. When exposed to Cd for a long time, the cellular structure and function of ovarian tissue or sperm tissue is significantly damaged, causing the reproductive ability of fish to reduce, the development of hatched embryos to be incomplete, and the growth and development of juvenile fish to be significantly retarded (Figure 1) [97,98]. Cd not only destroys the structural and functional integrity of fish gonads, but also affects the reproductive regulation system of fish by affecting the expression and secretion of hormones [99,100]. On the one hand, Cd can affect the expression and secretion of endocrine hormones, mainly by affecting the synthesis and secretion of hormones related to the hypothalamic–pituitary–gonadal (HPG) axis, and Cd also directly inhibits the expression of estrogen receptors [22,101]. For example, under Cd exposure, the expression of estrogen receptors in the gonads of *Oncorhynchus mykiss* was significantly inhibited, which affected the downstream pathways involved in estrogen receptors, resulting in disordered yolk production, abnormal egg development, and finally, reproductive abortion [102,103]. In addition, after Cd ion exposure, the pituitary gland secreted gonadotropin abnormally, resulting in the secretion of estrogen disorder and, finally, reducing the reproductive ability of fish. In addition to the direct inhibition of estrogen receptor expression, Cd can also bind to estrogen receptors in the form of estrogen analogs, thereby reducing the concentration of estrogen receptors in the gonads [31,102,104]. On the other hand, Cd can also affect the content of sex hormones in fish serum by inhibiting the synthesis of sterols, which has been demonstrated in mullet and guppies, in which the offspring numbers and survival rates were decreased under Cd stress conditions [103,105]. In addition, Cd can damage the genesis and maturation of germ cells, and high concentrations of Cd can reduce sperm motility and sperm number [106]. Zebrafish is a standard model animal for toxicity assessment, and studies showed that, after zebrafish were exposed to Cd solution, the gonad somatic index (GSI) of the ovary and seminal gland decreased significantly [18]. Cd mainly causes developmental abnormalities in zebrafish embryos, and the toxicity results of Cd in zebrafish are summarized in Table 2.

### 3.3. Cadmium Toxicity Damages the Immune System of Fish

The toxic effect of Cd on the fish immune system mainly occurs on the nonspecific immune system, and this affect mainly results in the inhibition of the number of white blood cells, the proliferation of lymphocytes, the phagocytosis rate of neutrophils, and the activity of macrophages [107,108]. The hemolymph system is an important target of Cd toxicity, and Cd exposure affects the organizational structural integrity of the fish hemolymph system and causes dysfunction, thereby resulting in an immunosuppressive effect [109,110]. Both in vitro and in vivo toxicity tests on *Ictalurus melas* showed that the proliferation of lymphocytes was significantly inhibited when the concentration of Cd used for exposure was higher than 2 μmol/L [111]. It has been found that exposure to high concentrations of Cd can inhibit the activities of lysozyme, alkaline phosphatase, and acid phosphatase in the liver of yellow catfish [112]. In addition, Cd can induce the proliferation of neutrophils in zebrafish larvae, and the secreted neutrophils will gradually accumulate in the inflammatory site of liver tissue (Figure 1) [113].

The other pathways that inhibit the nonspecific immune system, due to Cd exposure, inhibit the synthesis of essential trace elements [114]. Chronic exposure to environmental cadmium affects growth and survival, cellular stress, and glucose metabolism in juvenile channel catfish (*Ictalurus punctatus*) [115]. Because both zinc and copper are essential trace elements for organism immunity, if the zinc content is insufficient in an organism, the coefficient of the thymus, spleen, lymph node, and other organs will be reduced, and the cellular immune function will be reduced. Furthermore, if the copper content in the organism is deficient, humoral and cellular nonspecific immune functions will be reduced [114,116,117,118].

### 3.4. Cadmium Toxicity Damages the Energy Metabolism System of Fish

Energy metabolism plays an essential role in the growth and survival of organisms and is the basis of various life processes, such as proliferation and development, immune stress, and osmotic pressure regulation [19,119]. Cd can change ATPase activity and affect the function of ion channels. Studies have found that the activities of Ca^2+^-ATPase are inhibited as a result of the decrease in Ca^2+^ concentration in blood, which may be because Cd competitively binds to calcium channels in the gills after Cd exposure [120,121]. Researchers found that the activities of Na^+^/K^+^-ATPase in gills could be inhibited when the concentration of Cd in water was up to 1 mg/L, and the reason was that a large number of free radicals oxidize the amino acid and destroy the structure of ATPase, thus showing the inhibitory effects of Cd on ATPase [122]. Cd can not only affect the energy metabolism process in fish, but also cause damage to mitochondria, the main energy metabolism site. Regarding the effects of Cd on the mitochondria in *Silurus meridionalis*, it was shown that in *Silurus meridionalis* exposed to 500 μg/L Cd, the mitochondrial structure and function of the liver was significantly damaged and the activity of mitochondria enzymes was decreased, which results in a decrease in the energy of the liver metabolism [95]. Additional evidence verified that Cd could significantly reduce the level of ATP in the hemolymph of *Ostreidae* and induce energy metabolism disorder [123]. In addition, Cd exposure significantly affects the respiratory control rate and oxidative phosphorylation efficiency of mitochondria in *Ostreidae* and *Oncorhynchus mykiss* and interferes with the expression of key mitochondrial genes [53,124]. For example, Cd can induce the production of differentially expressed metabolites, such as succinate and glycogen, and proteins, including isocitrate dehydrogenase, pyruvate carboxylase, and glyceraldehyde-3-phosphate, that are related to energy metabolism in the gill tissues of *Tegillarca granosa* (Figure 1) [125].

### 3.5. Cadmium Toxicity Affects Nervous System Development in Fish

The neurotoxicity caused by Cd stress seriously threatens the normal life activities of fish because Cd directly inhibits the biological activities of enzymes that contain sulfhydryl by binding with sulfhydryl groups. At the same time, Cd leads to a decrease in the content of a series of metabolic intermediates, which causes serious damage to the nervous system of fish [126,127,128]. Cd can enter the brain tissue of fish and affect cerebellar functions, causing the fish to lose balance during swimming, while the effects on the brain are mainly neuronal damage. Cd exposure inhibited the formation of neural crest cells during the early development of fish, resulting in motor neuron damage and behavioral abnormalities [27,29,129]. Research has shown that Cd induces damage to nerve cells in the lateral line system of *Dicentrarchus labrax* and affects its escape and swimming behavior [130]. Cd also significantly inhibited acetylcholinesterase (AChE) activity in the brain tissue of zebrafish and *Lepisosteus oculatus* and induced neurological disorders [131,132]. Similarly, the behavior of juvenile zebrafish was severely inhibited under Cd exposure conditions [133]. In addition, Cd accumulates in the olfactory epithelium and bulb of fish, which reduces the expression of the genes related to olfactory sensory neurons, thereby damaging the olfactory sense of fish and damaging their anti-predator behavior (Figure 1) [134].

### 3.6. Cadmium Toxicity Leads to Changes in Blood Plasma Parameters

A study on *Nile tilapia* showed that albumin concentrations and serum total protein were decreased significantly (*p* < 0.001), while serum creatinine, glutamic-pyruvic transaminase (GPT), and aspartate aminotransferase (AST) significantly (*p* < 0.001) increased with increasing cadmium level in fish diets, and the urea-N concentration was insignificantly affected [135]. The results of the study on carp showed that cadmium poisoning would cause renal tubule reabsorption dysfunction and a large amount of calcium and phosphorus excretion, along with urine, and result in a decrease in serum calcium and phosphorus. In addition, cadmium can damage renal tubules, inhibit the activation of vitamin D, and also show a series of negative effects on calcium and phosphorus metabolism [92,136,137]. Usually, aminotransferases are most abundant in the liver and kidney, as well as in the heart muscle and bones, with very little in the serum, however, when these tissues, especially the liver and kidneys, are damaged or diseased, the enzyme enters the bloodstream and increases the activity of transaminase in the serum. Therefore, the increase and decrease of glutamic oxalacetic transaminase (GOT) and GPT activities in the serum may reflect the poisoning or pathological changes [138,139]. A study on the effects of cadmium exposure to *Sparus aurata* suggested that GOT activity decreased in liver cytoplasm, but increased in serum (Figure 1) [140].

## 4. Mechanism of the Toxic Effects of Cadmium Exposure on Fish

### 4.1. Cadmium Toxicity Leads to Oxidative Damage in Fish

When fish are stressed due to Cd in the water environment, a large number of reactive oxygen species (ROS) and reactive nitrogen species (RNS) radicals, which are beyond their own scavenging capacity, will accumulate in the fishs’ bodies, thereby resulting in a series of oxidative damage reactions [42,108]. The toxic effect of Cd on fish is mainly reflected in the oxidative stress caused by ROS, and the excessive accumulation of ROS will lead to structural changes in biological macromolecules, such as proteins and DNA, disorder of DNA replication and repair, and eventually pathological changes [141]. At the same time, ROS can cause unsaturated fatty acids on the cell membrane lipid peroxidation reaction to produce malondialdehyde (MDA) and cause cell membrane structural damage, in addition, MDA itself is a kind of toxic substance [94,142]. Combined, all of these factors will further affect the function of enzymes related to regulating cell proliferation and differentiation and apoptosis on the membrane, finally causing a series of functional disorders and damage to the body [94,143,144]. The increase in intracellular ROS caused by Cd mainly occurs through the following pathways: (I) Cd affects cellular respiration by inhibiting the respiratory transmission chain in the mitochondria, resulting in an increase in byproduct ROS content [145]; (II) Cd can increase ROS content by increasing the activity of nitric oxide synthase and the expression level of the related genes in the body [146]; (III) Cd can also remove Ca, Cu, Fe, and other elements from the binding proteins in the cell, thereby causing an increase in calcium, copper, iron, and other elements in the cytoplasm, resulting in oxidative stress and an increase in intracellular ROS content [147,148]; (IV) Cd reduces the antioxidant capacity of cells by reducing the activity of intracellular antioxidant enzymes (Figure 2) [143,149].

### 4.2. Cadmium Toxicity Affected the Expression of Stress Genes in Fish

Usually, when fish are subjected to various stresses in a water environment, a series of stress-related genes in the body are significantly inhibited or induced to be expressed, so that the toxic effect of stress can be alleviated through the body’s self-regulation [150]. As a stress defense mechanism, multiple types of genes are induced to be expressed under Cd stress, such as the oxidative stress-related genes *GR*, *CYP1A*, *MT*, and *HSPs*, antioxidant enzyme genes, such as *CAT*, *SOD*, and *GSH-Px*, and apoptosis-related genes, such as *Bax*, *Bcl-2*, *Caspase-3*, *Caspase-8*, and *Caspase-9* [143,151]. The expression levels of antioxidant enzyme genes in the liver, spleen, and muscle of *Takifugu obscurus* exposed to Cd in a water environment were detected, and the results showed that the expression levels of *MN-SOD*, *Cu/Zn-SOD*, and *CAT* genes in the liver and spleen were significantly increased. These levels in muscle tissue were inhibited at the early stage and significantly increased at the middle and late stages of Cd stress, indicating that the muscle produced a large amount of antioxidant enzymes to protect the fish body in the middle and late stages of Cd stress through body self-regulation [152]. Therefore, the fish body can regulate the generation of a large number of antioxidant enzymes to reduce the oxidative damage caused by ROS. Moreover, Cd also induces the abnormal expression of stress response protein genes, such as heat shock protein and metallic globulin, as determined in studies on the liver in *Cyprinus carpio* and *Labeo rohita*, in which the expression levels of *CYP1A*, *HSP47*, *HSP60*, *HSP70*, *HSP90*, and *MT-B* genes were significantly increased and then inhibited with prolonged Cd exposure time [153,154]. Furthermore, the expression levels of anti-stress genes in different tissues of *Cyprinus carpio* were significantly different after Cd exposure for 96 h. For example, the expression level of *HSP60* in the liver was significantly induced, but the expression levels of *HSP70* and *HSP90* were not significantly changed and inhibited. The expression level of *HSP70* in the gills first significantly increased and then decreased, and the *HSP90* expression level was significantly inhibited at all times (Figure 3) [153,155,156].

### 4.3. Cadmium Toxicity Inhibits Multiple Enzyme Activities

Cd can reduce the activities of many enzymes in fish, especially antioxidant enzymes that contain zinc and sulfhydryl, and Cd can bind to glutathione and metallothionein, which are two important thiol proteins that play an antioxidant role, thus, the ability of ROS elimination in the body is reduced and oxidative damage eventually occurs [157,158]. Cd binds to the sulfhydryl groups of superoxide dismutase (SOD) and glutathione reductase (GR), forms a Se-Cd complex with selenium (Se) in glutathione peroxidase (GPx), or replaces Zn in SOD to form Cd-SOD, thus, the activity of these enzymes is reduced or lost [159,160]. Studies have shown that the activities of SOD, catalase (CAT), and glutathione peroxidase (GPx) display an increasing trend in fish exposed to Cd at low concentrations, but usually show a decreasing trend at high concentrations, and the dose‒response curve of Cd generally shows a parabolic form, which has been verified in the liver and gills of *Charybdis japonica* [161]. Furthermore, excessive ROS will further lead to lipid peroxidation, mediate DNA damage, and promote the activation of poly-ADP-ribotransferase, thus reducing intracellular ATP levels. In addition, Cd inhibits the repair of DNA damage and the activity of DNA polymerase β, thereby affecting the immune function of the fish (Figure 3) [162,163].

## 5. Conclusions and Future Perspectives

This review summarizes the biological accumulation and toxicological effects of Cd in fish, including the research progress on the toxic effects of Cd on tissue structure and functional integrity, physiological metabolism, reproduction, immune system, and blood plasma parameters. We focus on the toxicological effects and mechanism of Cd in fish and anticipate future research directions to discover and reduce Cd pollution in the aquatic environment in a timely manner, achieve the purpose of protecting the aquatic ecological environment and aquatic life, and provide scientific evidence for humans to develop safe aquatic products.

At present, research on Cd toxicity is very active. Although many achievements have been made, from the molecular and metabolic perspectives, we believe that further research needs to be performed, as follows: (I) analyze the effects of low-dose and long-period Cd stress on fish and establish a database of toxicological effects of fish over time to provide a scientific basis for formulating water environmental sanitation standards; (II) strengthen the study on the combined toxicological effects of Cd and other environmental pollutants in the water environment, improve the database of toxicological indicators of the water environment and fish collected in the field, and comprehensively evaluate the ecotoxicological effects of Cd in the water environment; (III) and it is of great significance to strengthen the research on the toxicological effects and mechanisms of different forms of Cd on fish, to evaluate the ecological risk of Cd completely, to set environmental quality standards for Cd, and to repair and control Cd pollution in the water environment. Anyway, the mechanisms of the toxic effects of cadmium, combined with other heavy metals, on fish needs to be further studied. In recent years, the environmental hazards of microplastics are gaining researchers’ attention, and microplastics enhance the movement of cadmium through the food chain, which should be taken seriously.

## Figures and Tables

**Figure 1 toxics-10-00622-f001:**
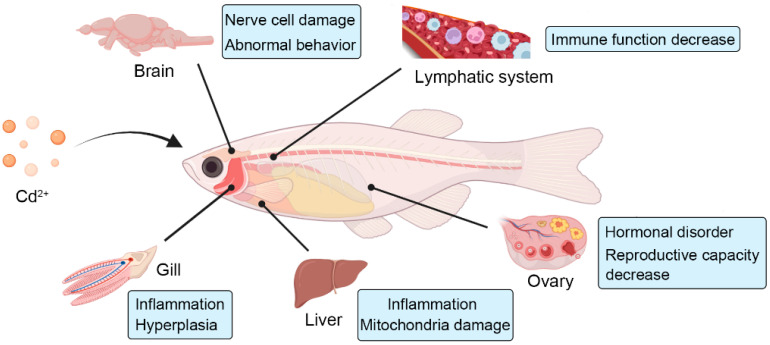
Schematic diagram of toxic effects of Cd on fish organs.

**Figure 2 toxics-10-00622-f002:**
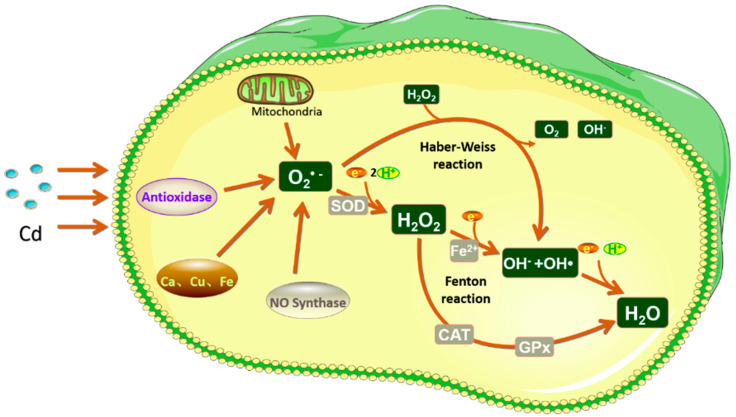
Schematic diagram of ROS metabolism and oxidative stress induced by Cd. Under normal physiological conditions, the ROS in the body are mainly converted into less toxic or harmless substances by antioxidant enzymes, such as superoxide dismutase (SOD), catalase (CAT), and glutathione peroxidase (GPx), while under the condition of Cd exposure, a large amount of ROS are generated and destroy the antioxidant system in cells.

**Figure 3 toxics-10-00622-f003:**
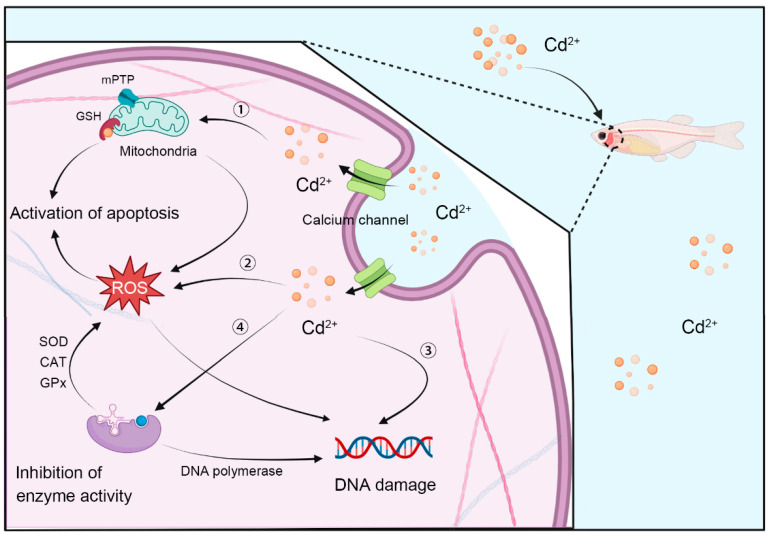
Schematic diagram of Cd toxicity on molecule and cell in fish.

**Table 1 toxics-10-00622-t001:** Reproductive and physiological endpoints for adult medaka pairs exposed to Cd for 7 weeks. Adapted with permission from [22]. Copyright 2003, Elsevier Inc. All rights reserved.

		Control	1 ppb Cd	5 ppb Cd	10 ppb Cd
Reproductive endpoints	Total eggs	99.2 ± 32.7	124.0 ± 29.5	134.7 ± 29.5	113.0 ± 26.1
	Total eggs/day	8.0 ± 2.0	9.2 ± 2.0	10.3 ± 1.4	8.6 ± 2.0
	Fertilized eggs/day	7.4 ± 1.8	8.9 ± 2.0	10.1 ± 1.3	8.3 ± 2.2
	Hatched eggs/day	6.2 ± 1.0	7.3 ± 0.9	9.4 ± 0.9*	7.6 ± 2.7
	Days to first hatch	12.4 ± 2.5	11.6 ± 1.5	10.5 ± 1.9	11.8 ± 2.3
	Percent survival	0.92 ± 0.4	0.88 ± 0.19	0.93 ± 0.04	0.98 ± 0.02
	Spawning frequency	12.2 ± 1.5	13.4 ± 0.5	13.0 ± 1.4	13.2 ± 0.7
	Egg size (μm)	1.34 ± 0.02	1.35 ± 0.02	1.34 ± 0.03	1.36 ± 0.02
Physiological endpoints (females)	Plasma VTG (IOD/positve)	0.35 ± 0.19	0.40 ± 0.20	0.38 ± 0.10	0.40 ± 0.15
	Hepatic ER (IOD/positive)	0.79 ± 0.10	0.72 ± 0.07	0.74 ± 0.06	0.74 ± 0.07
	GSI (mm^2^/mg)	0.050 ± 0.005	0.052 ± 0.008	0.051 ± 0.011	0.049 ± 0.003
	Plasma E2 (pg/μL)	29.2 ± 13.9	36.6 ± 4.5	54.3 ± 7.1 *	12.4 ± 5.9 *
	Plasma T (pg/μL)	10.3 ± 6.8	6.5 ± 2.8	15.9 ± 3.6	18.4 ± 25.1
	Gonadal E2 (pg)	314 ± 90	58 ± 63 *	56 ± 50 *	164 ± 108 *
	Gonadal T (pg)	561 ± 106	261 ± 146 *	167 ± 168 *	121 ± 62 *
Physiological endpoints (males)	Plasma VTG (IOD/positve)	0.18 ± 0.02	0.16 ± 0.01	0.17 ± 0.01	0.19 ± 0.01
	Hepatic ER (IOD/positive)	0.78 ± 0.15	0.73 ± 0.05	0.76 ± 0.05	0.76 ± 0.08
	GSI (mm^2^/mg)	0.013 ± 0.002	0.011 ± 0.002	0.007 ± 0.003 *	0.010 ± 0.003
	Plasma E2 (pg/μL)	39.1 ± 15.3	62.0 ± 17.1	65.6 ± 4.4	33.0 ± 10.4
	Plasma T (pg/μL)	7.4 ± 4.7	11.9 ± 2.0	9.7 ± 3.4	28.4 ± 36.8
	Gonadal E2 (pg)	638 ± 219	BD *	BD *	89 ± 154 *
	Gonadal T (pg)	326 ± 109	168 ± 98 *	88 ± 57 *	115 ± 57 *

Values are mean S.D. VTG, vitellogenin; ER, estrogen receptor; GSI, gonadal somatic index; E2, estradiol; T, testosterone.* Significantly different at *p <* 0.05 from the corresponding control value for male and female medaka. BD is below detection limit.

**Table 2 toxics-10-00622-t002:** Summary of toxic effects of cadmium in zebrafish. Reproduced with permission from [25]. Copyright 2021, Korean Society of Toxicology.

Toxicity		Ref.
**Embryos**		
Liver	Hepatic lipid accumulation	[26]
Nerve	Neuroglia alterations	[27]
	Increased ATPase activity in brain	[28]
	Reduction of neuronal differentiation and axonogenesis	[29]
	Interference of neural development	[30]
	Anti-estrogen in brain	[31]
	Abnormal somite patterning	[32]
Myoskeletal retina	Eye hypoplasia and hypopigmentation	[33]
Cardiovascular organ	Heart edema and increased pericardial area	[34]
	Activation of cell death pathway in olfactory epithelium	[35]
Olfactory organ	Delay in hatching time	[34]
Others	Tail and axis malformation	[34]
**Larvae**		
Nerve	Circadian rhythms disruption	[36]
Others	Cell death and structural alterations in olfactory epithelium	[37]
**Adults**		
Liver	Carcinogenesis	[38]
	Hepatic lipid accumulation	[39]
	Oxidative damage	[40,41]
Nerve	Oxidative damage	[41,42]
Myoskeletal	Structural disorganization, disassembly of muscular myofibrils	[43]
Reproductive organ	Pair spawning reduction and teratogenicity	[44]
	Ovary: oxidative damage	[41]
Retina	Nerve fiber thickening and vacuolating	[45]

**Table 3 toxics-10-00622-t003:** Table of lethal effects of cadmium on different fish species.

Species	Concentration	Time	Effect	Ref.
*Silurus meridionalis*	6.85 mg/L	96 h	Median lethal	[58]
*Danio rerio*	25 μg/L	9 d	Median lethal	[59]
*Oryzias javanicus*	1.0 ppm	-	Embryo developmental arrest	[60]
*Lutjanus peru*	0.05 mM	2 h	Cell viability reduction	[61]
*Oncorhynchus mykiss*	8 µg /L	96 h	Embryonic mortality rate (97.5%)	[62]
*Rasbora sumatrana*	0.1 mg/L	96 h	Median lethal	[63]
*Poeciliareticulata*	0.17 mg/L	96 h	Median lethal	[63]
*Tautogolabrus adspersus*	26 μg/mL	96 h	Median lethal	[64]
*Morone saxatilis*	20 μg/mL	96 h	Median lethal	[64]
*Oreochromis niloticus*	14.8 mg/L	96 h	Median lethal	[65]
*Cyprinus carpio*	0.20 ± 0.16 μM	96 h	Median lethal	[66]
*Clarias gariepinus*	10.85 mg/L	96 h	Median lethal	[67]
*Labeo rohita*	89.5 mg/L	96 h	Median lethal	[68]
*Trichogaster (Colisa) fasciata*	49.5 mg/L	96 h	Median lethal	[69]
*Silurus soldatovi*	2.74 mg/L	96 h	Median lethal	[70]

**Table 4 toxics-10-00622-t004:** Minimum, maximum, mean, and standard deviation (SD) of Cd in canned tuna samples and fresh fish samples (mg/kg ww). Adapted with permission from [84]. Copyright 2019, Health Scope.

Species	Cd		
	Min	Max	Mean ± SD
Mako shark (*Isurusoxyrinchus*)	0.001	0.003	0.0023 ± 0.0005
Yellowfin tuna (*Thunnusalbacares*)	0.001	0.002	0.0019 ± 0.0001
Soupfin shark (*Galeorhinusgaleus*)	0.001	0.002	0.0018 ± 0.0002
Swordfish (*Xiphias gladius*)	0.001	0.003	0.0022 ± 0.0004

Abbreviation: ww, wet weight.

## Data Availability

Not applicable.

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
