# Peer review of "Toxic Effects of Cadmium on Fish"

_toxics, 2022, doi:10.3390/toxics10100622_

Round 1
Reviewer 1 Report
The authors of the manuscript entitled “ Toxic effects of cadmium on fish” aimed to review the toxic effects of cadmium (Cd) in fish. Follow my considerations and concerns:
Minor review:
- insert dot at line 18.
- italic style at line 178; other species too
- describe name’s specie, it was the first time appeared at line191
- unify terminology (CdCl2) at line 204
Major review:
In order to improve the article, I suggest to authors insert a summary table comparing Cd dose in embryos and adult fishes, in addition the major tocixicity roles of Cd, if Cd was administrated or in water, etc.
Moreover, the figures added in the manuscript weren’t related to the title and/or the scope of the review. I strongly suggest insert a figure summarized the effects of Cd in fish tissue and join figures 1 and 2 at the same figure.
Reviewer 2 Report
Throughout the review, the authors need to accurate the species of fish. Because different species leads to different responses.
Abstract
- The authors have many punctuation errors: missing final points and the authors need to replace all ; by ,
- Please provide new keywords, Cadmium exposure and fish are already in the title.
The Introduction is very imprecise, this need to be rewrite and more specific regarding aquatic organisms, with the suport of bibliography.
- Line 23 to 25: Cadmium (Cd), a toxic and nonessential transition metal, is a kind of environmental endocrine that is a major threat to common organisms in aquatic ecosystems and has been widely recognized as a major pollutant in waters worldwide. Cadmium have endocrine effect? Cadmium is an heavy metal, not an endcocrine disrupter.
_ Line 26, 27: In humans? Please specify. The review is regarding fish, not humans.
Future perspectives, need to be rewrite. The authors need to do an effort to to draw new conclusions from the combination of bibliography referred throughout the review.
Reviewer 3 Report
The Authors submitted a review concerning toxic effects of cadmium to fish. In the literature there is a vast amount of information concerning this subject, while the Authors used this literature rather selectively. I suggest to add a chapter concerning lethal values of Cd (e.g. limited to 96hLC50 which is the most commonly used measure) to various fish species (with a table containing fish species, Cd concentration and reference). I also suggest to add chapters on developmental: embryonic and larval toxicity and hematotoxicity since hematological parameters are widely used in fish toxicology. I suggest a considerable language revision. Particularly I do not like the word "enrichment" used by the Authors many times and suggest to use "accumulation" instead. Mechanisms of Cd absorption are decribed in the chapter 4, while thay should be presented at the begining of the chapter 2 or as a separate chapter showing the routes of Cd uptake (branchial and intestinal) and then - methods of Cd measurement. I think that sublethal effects could be shown in a table containing: fish species, Cd concentration, time of exposure, description of the effect and reference. Clear conclusion about toxic effects of Cd to fish should be formulated before the Future perspectives.
Round 2
Reviewer 1 Report
The authors revised the main points required by the reviewer and the manuscript improved its quality. The article can be accepted for publication.
Author Response
Thank you very much for your valuable suggestions on this manuscript.
Reviewer 2 Report
The authors made an effort to improve the manuscript. All the comments were answer.
Author Response

(The authors gave the same response as above.)

Reviewer 3 Report
The Authors considerably revised the manuscript, however, I again suggest to add a table with lethal values (there is a lot of literature available) and arrange the tables with sublethal effects in a different way: show species, concentration, effect and reference.
Round 3
Reviewer 3 Report
The manuscript was revised and in my opinion is ready to be published after a minor editorial and language revision.